Glomeromycota associations with bamboos (Bambusoideae) worldwide, a qualitative systematic review of a promising symbiosis

Sánchez-Matiz Juan José
Díaz-Ariza Lucia Ana luciaana@javeriana.edu.co
1 Grupo de Investigación en Agricultura Biológica, Laboratorio de Asociaciones Suelo Planta Microorganismo, Departamento de Biología/Facultad de Ciencias, Pontificia Universidad Javeriana , Bogotá , DC , Colombia
Adhikari Tika
Electronic publication date: 2023 Nov 9
Publication date: 2023
Volume: 11
Electronic Location ID: e16151
Received 2023 May 23; Accepted 2023 Aug 30
Copyright: ©2023 Sánchez-Matiz and Díaz-Ariza
Copyright year: 2023
Copyright holder: Sánchez-Matiz and Díaz-Ariza
License: This is an open access article distributed under the terms of the Creative Commons Attribution License, which permits unrestricted use, distribution, reproduction and adaptation in any medium and for any purpose provided that it is properly attributed. For attribution, the original author(s), title, publication source (PeerJ) and either DOI or URL of the article must be cited.
License URL: https://creativecommons.org/licenses/by/4.0/

Keywords: Glomeromycota, Arbuscular mycorrhizal fungi, Bambusoideae, Bamboo, Symbiosis

Funding: Ministerio de Ciencia, Tecnología e Innovación; Ministerio de Educación Nacional; Ministerio de Industria, Comercio y Turismo; ICETEX 792-2017 2a Convocatoria Ecosistema Científico-Colombia Científica para la Financiación de proyectos de I + D + i Banco Mundial, Vicerrectoría de Investigaciones, Pontificia Universidad Javeriana, Bogotá, Colombia Contrato Número FP44842-221-2018 Financial support for this study was provided bay a grant from Pontificia Universidad Javeriana, Ministerio de Ciencia, Tecnología e Innovación, Ministerio de Educación Nacional, Ministerio de Industria, Comercio y Turismo e ICETEX, 2a Convocatoria Ecosistema científico—Colombia Científica 792-2017, Programa “Generación de alternativas terapéuticas en cáncer a partir de plantas a través de procesos de investigación y desarrollo traslacional, articulados en sistemas de valor sostenibles ambiental y económicamente” (Contrat no. FP44842-221-2018). The funders had no role in study design, data collection and analysis, decision to publish, or preparation of the manuscript.

==============================
Background

Around the world, bamboos are ecologically, economically, and culturally important plants, particularly in tropical regions of Asia, America, and Africa. The association of this plant group with arbuscular mycorrhizal fungi belonging to the phylum Glomeromycota is still a poorly studied field, which limits understanding of the reported ecological and physiological benefits for the plant, fungus, soil, and ecosystems under this symbiosis relationship.

Methods

Through a qualitative systematic review following the PRISMA framework for the collection, synthesis, and reporting of evidence, this paper presents a compilation of the research conducted on the biology and ecology of the symbiotic relationship between Glomeromycota and Bambusoideae from around the world. This review is based on academic databases enriched with documents retrieved using different online databases and the Google Scholar search engine.

Results

The literature search yielded over 6,000 publications, from which 18 studies were included in the present review after a process of selection and validation. The information gathered from the publications included over 25 bamboo species and nine Glomeromycota genera from eight families, distributed across five countries on two continents.

Conclusion

This review presents the current state of knowledge regarding the symbiosis between Glomeromycota and Bambusoideae, while reflecting on the challenges and scarcity of research on this promising association found across the world.

Introduction

Bamboos belong to the subfamily Bambusoideae within the Poaceae family (commonly known as grasses) (Clark, Londoño & Ruiz-Sanchez, 2015) and tend to be evergreen plants with non-seasonal flowering, after which some species die (Clark, 1990; Banik, 2015; Liu et al., 2017). They are a highly versatile group of fast-growing plants for which more than 4,000 traditional uses and 1,500 commercial applications have been reported (Hsiung, 1988) and are currently used as fuel, building material, raw material for the paper industry, and even as a resource for creating artisanal crafts (Singh et al., 2020). These reasons, along with the resistance properties of their fibers, have earned them the nickname “vegetable steel” (Amada et al., 1997).

Furthermore, due to their phenotypic plasticity, bamboo plants are extensively cultivated in numerous regions around the world—in 2007, 36.8 million hectares were counted across temperate, tropical, and subtropical regions (Lobovikov et al., 2007). They have also come to be considered invasive plants in some temperate and tropical regions, such as China and Japan, since several species have leptomorphic or monopodial rhizomes commonly called “corridors” that have a high capacity for underground expansion (e.g., Phyllostachys spp.). On the other hand, pachymorphic or sympodial bamboos are more common in tropical and subtropical regions and have a limited expansion capacity (e.g., Guadua spp. or Bambusa spp.) (Buziquia et al., 2019; Xu et al., 2020).

The areas where bamboo species are most cultivated and consumed are concentrated in Asia, Africa and America (Bystriakova, Kapos & Lysenko, 2004; de Moura et al., 2019), where there is also greater diversity (de Moura et al., 2019). According to Soreng et al. (2022), there are around 136 genera and 1.698 species of bamboo distributed in the aforementioned zones, with tropical Asia considered the center of bamboo diversity, harboring around 53 genera and 550 species (Bystriakova, Kapos & Lysenko, 2003).

Plants do not exist as isolated entities but as complex communities where their organs and tissues constitute niches for diverse microorganisms (Kothe & Turnau, 2018). Among the multiple interactions between plants and organisms, those established in the phylum Glomeromycota stand out (Gehring & Johnson, 2017). This phylum comprises a monophyletic clade of fungi whose members except for the species Geosiphon pyriformis (Kütz). F. West.) are all obligate mutualistic symbionts of plants, better known as arbuscular mycorrhizal fungi or AMF (Smith & Read, 2008).

These fungal symbionts, which include over 350 described and accepted morphological species, establish associations with the roots of the vast majority of terrestrial plants around the world. According to recent estimates, this value surpasses 60% or even 80% of plant species on the planet (van der Heijden et al., 2015; Smith & Read, 2008; Prasad et al., 2017; Brundrett & Tedersoo, 2018), with the exceptions being a few plant families that do not exhibit any type of association, such as Amaranthaceae, Brassicaceae, Chenopodiaceae, Caryophyllaceae, Juncaceae, Cyperaceae, and Polygonaceae (Brundrett, 2009).

Once they colonize the roots of the host plant, AMF can develop extensive extraradical networks of mycelium that grow three-dimensionally in the soil matrix and specialize in capturing mineral nutrients and water. These nutrients and water are subsequently transported and translocated to the interior of the plant symbiont in exchange for an energy reward for the AMF, mainly in the form of carbohydrates, which are produced through photosynthesis (Smith & Read, 2008).

Of all the microorganisms present in soils, Glomeromycota fungi are fundamental to the maintenance and functionality of numerous ecosystem processes. Their symbiotic establishment is related to the development and growth of the plants with which they associate, the maintenance of plant diversity, nutrient cycling, phosphorus solubilization, the facilitation of water and nutrient capture, and soil aggregation, among other ecosystem contributions (Marulanda, Azcon & Ruiz-Lozano, 2003; Silva-Flores et al., 2019; van der Heijden et al., 2015; Fall et al., 2022). The importance of the association between AMF with Bambusoideae, specifically, has been highlighted through the properties and effects described in terms of contributions to growth and nutrient capture, and even as an important component of soil respiration in the C cycling of the bamboo forest ecosystem, among others (Babu & Reddy, 2010; Muthukumar & Udaiyan, 2006; Jha et al., 2011; Jin et al., 2022).

This group of plants, once considered the “wood of the poor,“ is now recognized as “green gold” and presents particularly promising perspectives for environmental issues and a rapidly growing market (Sandhu & Wani SH, 2017). Thus, the development of strategies is required to propagate bamboo species quickly and economically, and the stimulation of their growth and development can be achieved through the use of beneficial microorganisms such as AMF (Zamora-Chacón et al., 2019; de Moura et al., 2019). Therefore, the objective of this systematic review is to provide a summary of the current state of qualitative knowledge reported in the global scientific literature regarding the symbiosis between Glomeromycota fungi and bamboo plants of the Bambusoideae subfamily with which they associate worldwide.

Materials & Methods

The present study utilizes a literature search and review process that is summarized in the PRISMA (Preferred Reporting Items for Systematic Reviews and Meta-Analyses) flowchart, adapted from Page et al. (2021) and Moher et al. (2009). This flowchart follows a clear separation of the stages of identification, screening, eligibility, and inclusion of literature (Fig. 1). An updated PRISMA checklist (Page et al., 2021) with a guide specific to this review is presented in (Table S1).

Figure 1 Workflow displaying the search, selection, and eligibility criteria applied to the literature, adapted from the PRISMA 2020 guidelines for reporting new systematic reviews (Page et al., 2021).

Search strategies and inclusion criteria

To gather relevant literature on the symbiotic association between Glomeromycota and bamboo plants (Bambusoideae) worldwide, academic and bibliographic databases including Scopus, Web of Science, Wiley Online Library, Taylor and Francis Online, Elsevier Science Direct, and Springer Link, as well as the search engine Google Scholar, were utilized. Documents published up until the search date of February 2023 were recovered using the following search equation with keywords and Boolean operators: (Glomeromycota OR “arbuscular mycorrhizal fungi” OR arbuscular OR mycorrhiza OR AMF) AND (Bambusoideae OR Bamboo) AND (association OR symbiosis).

The search was conducted and assessed by both authors using both English and Spanish terms, without imposing language restrictions or applying time restrictions in terms of the date of publication to avoid potential bias. This initial search resulted in over 6,300 publications, from which scientific articles published in peer-reviewed and indexed journals were selected, while books, book chapters, and reviews were excluded. The reference list of the selected literature was used to include additional articles, resulting in a total of 18 scientific articles included in the final review (Fig. 1).

Data extraction, compilation, and exclusion criteria

The following information related to the focal association was extracted from the compiled selected articles: (1) the country where the study was conducted, (2) the compartment used to perform the taxonomic identification of Glomeromycota specimens, whether it was the soil associated with bamboo plants, bamboo roots, or both; (3) the taxonomic genus of the bamboo host of the AMF, (4) the species (and, if known, variety) of bamboo hosting the AMF; (5) the family, (6) genus, and (7) species (if identified) of the Glomeromycota fungi associated with bamboo, as well as (8) the type of identification used to determine the taxonomic categories of the AMF found, whether it was morphological or molecular (i.e., based on phylogenetic markers).

From these documents, secondary information related to the symbiosis established between Glomeromycota and Bambusoideae was also extracted and discussed, particularly information that referred to the effects of the establishment of symbiosis between these groups of fungi and plants. As such, out of the total number of publications found and selected, those that did not explicitly state the taxonomic identity (at any level) of the plant and its associated arbuscular mycorrhizal fungus, beyond the generic classifications of “bamboos” and “Glomeromycota,” were excluded from the meta-synthesis, as well as publications classified as “grey literature”.

Lastly, while the taxonomic information extracted from the mentioned documents was not modified, it was verified according to the latest updates in the mycological databases Index Fungorum (http://www.indexfungorum.org/ and MycoBank (http://www.mycobank.org/ for the Glomeromycota fungi, and according to the freely accessible book “World Checklist of Bamboos and Rattans” for the bamboo plants (Vorontsova et al., 2016).

Results

Worldwide studies on AMF-bamboo association

In the field of mycorrhizal symbiosis established between Glomeromycota and Bambusoideae, most studies have been carried out in Asia, particularly in India (Bhattacharya et al., 2002; Debnath et al., 2015; Jamaluddin & Turvedi, 1997; Das et al, 2021; Das & Kayang, 2010; Jha et al., 2011; Parkash, Handique & Dhungana, 2019; Babu & Reddy, 2010; Ravikumar et al., 1997; Verma & Arya, 1998; Muthukumar & Udaiyan, 2006; Singh et al., 2020) and China (Guo et al., 2023; Xing et al., 2021; Qin et al., 2017a; Qin et al., 2017b; Jin et al., 2022; Zhang et al., 2022; Zhang et al., 2019; Weixin, Guangqian & Yulong, 2013). There are also some studies from Indonesia (Mansir et al., 2021; Kramadibrata, Prastyo & Gunawan, 2007; Kramadibrata, 2011) and one each from Sri Lanka and Japan (Mafaziya, Wijewickrama & Madawala, 2019; Fukuchi et al., 2011). However, from the Americas, only one report has been published by an indexed journal, a study carried out in Brazil (de Moura et al., 2019). No studies have been reported in other countries or regions where bamboo is currently cultivated and develops naturally, as in the case of the genus Guadua, whose distribution and usage spans from Mexico to Argentina in Neotropical America (Long, Yanxia & Jayaraman, 2022). Nevertheless, not all the studies mentioned were included in the review for the reasons previously described, leading to a reduced number of publications presented in Fig. 2, which summarizes how many publications were carried out in each country. Bambusoideae taxa reported from each country and the literature references are presented in Table 1, and the taxonomic groups of Glomeromycota reported in the selected studies from each country are presented in Table 2.

Figure 2 Map showing the number of studies from each country that include symbiotic associations between Glomeromycota and bamboos and identify both symbionts at least to the taxonomic level of the family.

Table 1 Countries from which bamboo genera and/or species associated with Glomeromycota have been reported.

Bambusoideae taxa	Countries where reported	References	
	Brazil	China	India	Indonesia	Sri Lanka		
Actinocladum	+						
A. verticillatum	+					8	
Bambusa	+	+	+	+	+		
B. arundinacea			+			1	
B. bambos			+		+	2,3,1,4	
B. blumeana				+		5	
B. burmanica			+			1	
B. grandis		+				6	
B. nana			+			1	
B. nutans			+			2,1	
B. pallida			+			2	
B. pervariabilis		+				6	
B. polymorpha			+			1	
B. tulda			+			7,2	
B. vulgaris	+		+	+		8,1,9	
B. sp.				+		10	
Cephalostachyum			+				
C. pergracile			+			1	
Dendrocalamus			+	+			
D. asper			+	+		11,5,9,1	
D. hamiltonii			+			7	
D. hookeri			+			7	
D. membranaceus			+			1	
D. strictus			+			14,15,4,1,13,12	
Dinochloa				+			
D. sp.				+		5	
Gigantochloa				+			
G. apus				+		9	
G. atter				+		5,16	
G. manggong				+		9	
Melocanna			+				
M. baccifera			+			1	
Nastus				+			
N. reholttumianus				+		5	
Phyllostachys		+	+				
P. edulis		+				17,18	
P. mannii			+			7	
Schizostachyum				+			
S. brachycladum				+		5	
S. lima				+		5	
S. zollingeri				+		9	
Notes.

1 Jamaluddin & Turvedi (1997) 2 Jha et al. (2011), 3 Mafaziya, Wijewickrama & Madawala (2019), 4 Parkash, Handique & Dhungana (2019), 5 Kramadibrata (2011), 6 Weixin, Guangqian & Yulong (2013), 7 Das & Kayang (2010), 8 ((de Moura et al., 2019)), 9 Kramadibrata, Prastyo & Gunawan (2007), 10 Husein et al. (2022), 11 Verma & Arya (1998), 12 Ravikumar et al. (1997), 13 Muthukumar & Udaiyan (2006), 14 Bhattacharya et al. (2002), 15 Babu & Reddy (2010), 16 Mansir et al. (2021), 17 Qin et al. (2017a), 18 Qin et al. (2017b).

Table 2 Countries from which Glomeromycota families and/or genera associated with bamboo (Bambusoideae) have been reported, along with the specific reference of the report of each fungal taxa.

Glomeromycota taxa and associated reference(s)	Countries where AMF species have been reported with bamboo	
	Brazil	China	India	Indonesia	Sri Lanka	
Acaulosporaceae	+	+	+	+	+	
Acaulospora	+		+	+	+	
de Moura et al. (2019)	+					
Husein et al. (2022)				+		
Jamaluddin & Turvedi (1997)			+			
Jha et al. (2011)			+			
Kramadibrata, Prastyo & Gunawan (2007)				+		
Kramadibrata (2011)				+		
Mafaziya, Wijewickrama & Madawala (2019)					+	
Mansir et al. (2021)				+		
Das & Kayang (2010)			+			
Parkash, Handique & Dhungana (2019)			+			
Verma & Arya (1998)			+			
Entrophospora			+			
Parkash, Handique & Dhungana (2019)			+			
Ni		+				
Qin et al. (2017a)		+				
Qin et al. (2017b)		+				
Ambisporaceae		+	+			
Ambispora			+			
Das & Kayang (2010)			+			
Ni		+				
Qin et al. (2017a)		+				
Qin et al. (2017b)		+				
Archaeosporaceae		+				
Ni		+				
Qin et al. (2017a)		+				
Qin et al. (2017b)		+				
Claroideoglomeraceae	+	+				
Claroideoglomus	+					
de Moura et al. (2019)	+					
Ni		+				
Qin et al. (2017a)		+				
Qin et al. (2017b)		+				
Diversisporaceae	+	+				
Diversispora	+					
de Moura et al. (2019)	+					
Ni		+				
Qin et al. (2017a)		+				
Qin et al. (2017b)		+				
Gigasporaceae	+	+	+	+	+	
Gigaspora	+		+	+	+	
Bhattacharya et al. (2002)			+			
de Moura et al. (2019)	+					
Husein et al. (2022)				+		
Jamaluddin & Turvedi (1997)			+			
Glomeromycota taxa and associated reference(s)	Countries where AMF species have been reported with bamboo	
	Brazil	China	India	Indonesia	Sri Lanka	
Mansir et al. (2021)				+		
Parkash, Handique & Dhungana (2019)			+			
Verma & Arya (1998)			+			
Ni		+				
Qin et al. (2017a)		+				
Qin et al. (2017b)		+				
Scutellospora	+		+	+	+	
Babu & Reddy (2010)			+			
de Moura et al. (2019)	+					
Jamaluddin & Turvedi (1997)			+			
Kramadibrata, Prastyo & Gunawan (2007)				+		
Mafaziya, Wijewickrama & Madawala (2019)					+	
Parkash, Handique & Dhungana (2019)			+			
Verma & Arya (1998)			+			
Glomeraceae	+	+	+	+	+	
Glomus	+	+	+	+	+	
Babu & Reddy (2010)			+			
Bhattacharya et al. (2002)			+			
de Moura et al. (2019)	+					
Husein et al. (2022)				+		
Jamaluddin & Turvedi (1997)			+			
Jha et al. (2011)			+			
Kramadibrata, Prastyo & Gunawan (2007)				+		
Kramadibrata (2011)				+		
Mafaziya, Wijewickrama & Madawala (2019)					+	
Mansir et al. (2021)				+		
Muthukumar & Udaiyan (2006)			+			
Das & Kayang (2010)			+			
Parkash, Handique & Dhungana (2019)			+			
Ravikumar et al. (1997)			+			
Verma & Arya (1998)			+			
Weixin, Guangqian & Yulong (2013)		+				
Sclerocystis	+		+			
de Moura et al. (2019)	+					
Jamaluddin & Turvedi (1997)			+			
Ni		+				
Qin et al. (2017a)		+				
Qin et al. (2017b)		+				
Paraglomeraceae		+				
Ni		+				
Qin et al. (2017a)		+				
Qin et al. (2017b)		+				
Notes.

Ni, (Not identified).

Categorization and description of the general framework of the studies

In these studies, some of the most common approaches involved analyzing rhizospheric and non-rhizospheric soils in search of AMF spores. From this. morphological evaluations were carried out that resulted in genus-level resolution in most cases, with some studies being able to identify further to the species level. A considerably small percentage of studies applied molecular approaches. Such studies used BLAST and specific fungal sequence databases to search for genetic similarities with previously described species. Other studies included an analysis of host bamboo roots that often calculated the percentage of mycorrhization or root colonization by AMF and related this to edaphic or biotic variables such as soil fertility and seedling dry weight in bamboo experiments, respectively. In another study, through an analysis of rhizospheric soils from several species of bamboo and their microbial communities, (Xing et al., 2021) found that at the phylum level, the relative abundance of Glomeromycota was higher in Phyllostachys edulis than in other species of bamboo such as Phyllostachys sulphurea, Phyllostachys bambusoides, Sinobambusa tootsik and Sasa auricoma. Meanwhile, (Jamaluddin & Turvedi, 1997) planted a bambusetum in boreholes with sand, FYM, and silt soil under natural conditions in a basaltic landscape of India with thirteen different bamboo species, corresponding to Bambusa vulgaris var. vittata and var. striata, Bambusa nutans, Bambusa nana, Bambusa bambos, Bambusa arundinacea, Bambusa burmanica, Bambusa polymorpha, Cephalostachyum pergracile, Dendrocalamus asper, Dendrocalamus strictus, Dendrocalamus membranaceus and Melocanna baccifera, and after a subsequent evaluation of their roots, reported that all of them established associations with AMF. Another study included Bambusa bambos from three tropical moist evergreen forests from Sri Lanka, and all presented the presence of AMF in their rhizospheric soils (Mafaziya, Wijewickrama & Madawala, 2019). Kramadibrata, Prastyo & Gunawan (2007) studied soil samples from several species of bamboo in Java, Indonesia, and Kramadibrata (2011) described AMF from soils associated with eight species of bamboo on the island of Sumba, Indonesia. More recently, de Moura et al. (2019) characterized the AMF community associated with the bamboo species Actinocladum verticillatum and Bambusa vulgaris var. vittata in Brazil, which showed no significant differences in colonization rates among the two plant species. Also, Das & Kayang (2010) described that Bambusa tulda exhibits an Arum-type AMF colonization, while other bamboo species (Dendrocalamus hookeri, Dendrocalamus hamiltonii, and Phyllostachys mannii) present Paris type colonization, indicating a difference in the way the plant and its fungal symbionts interact at the genetic level (Dickson, 2004; Cuba et al., 2020).

Another case (Fukuchi et al., 2011) described how the Japanese bamboo species Sasa senanensis establishes arbuscular mycorrhizal associations with Glomeromycota; however, they did not report the fungal taxonomy. Debnath et al. (2015) evaluated the presence of AMF in the roots of several bamboo species from two sampling sites in India. The first sampling site included the species Bambusa balcooa, B. tulda, B. bambos, B. cacherensis, B. tuldoides, Dendrocalamus hamiltonii, D. asper, and Oxytenanthera nigrociliata, while the second contained only two species, Bambusa vulgaris, and Bambusa polymorpha. All species from both sites presented roots colonized by AMF. It has also been noted that the bamboo species Dendrocalamus strictus is moderately sensitive to AMF colonization, with the presence of such association being restricted to lateral roots, particularly those of the third and second order, respectively, but rarely in the first order roots (Bhattacharya et al., 2002).

Based on the aforementioned considerations and information available from each study, the primary factor in categorizing these studies was the type of tools used for the identification of Glomeromycota species or taxonomic groups associated with each bamboo species or group, whether through molecular tools, morphological tools, or both (an event that was not observed). The secondary categorization of these studies involved the compartment type used for such taxonomic identification, which could vary depending on the identification technique, since molecular techniques are not restricted by the type of compartment (roots or soil), while morphological techniques can only explore spores extracted from the soil given that exact identification within the roots is practically impossible (Smith & Read, 2008). This categorization allows for an assessment of tools used and compartments analyzed per Bambusoideae species or groups, as shown in Table 3, where it is evident that the only compartment analyzed in all studies, regardless of the AMF identification technique, was soil. It is also apparent that the molecular identification of groups within Glomeromycota was only performed in two studies, both on the same bamboo species, Phyllostachys edulis.

Table 3 Identification method and compartment type from which Glomeromycota have been reported in different bamboo species, as well as the fungal groups described for each bamboo at the genus and family levels.

Compartment and Bambusoideae taxa	Glomeromycota taxa	Ref.	
	Ac	Am	Ar	Cl	Di	Gi	Gl	Pa		
	Aca	Ent	Ni	Amb	Ni	Ni	Cla	Ni	Div	Ni	Gig	Scu	Ni	Glo	Scl	Ni	Ni		
Molecular (Soil)			+		+	+		+		+			+			+	+		
Phyllostachys			+		+	+		+		+			+			+	+		
P. edulis			+		+	+		+		+			+			+	+	17,18	
Morphological (Soil)	+	+		+			+		+		+	+		+	+				
Actinocladum	+						+		+		+	+		+	+				
A. verticillatum	+						+		+		+	+		+	+			8	
Bambusa	+	+					+		+		+	+		+	+				
B. arundinacea											+	+		+	+			1	
B. bambos	+	+									+	+		+	+			1,2,3,4	
B. blumeana	+																	5	
B. burmanica											+	+		+	+			1	
B. grandis														+				6	
B. nana	+										+	+		+	+			1	
B. nutans	+										+			+	+			1,4	
B. pallida	+													+				4	
B. pervariabilis														+				6	
B. polymorpha	+													+	+			1	
B. sp.	+										+			+				10	
B. tulda	+											+		+				7,4	
B. vulgaris	+						+		+		+	+		+	+			8,1,9	
Cephalostachyum											+	+		+	+				
C. pergracile											+	+		+	+			1	
Dendrocalamus	+			+							+	+		+	+				
D. asper	+										+	+		+				1,9,5,11	
D. hamiltonii	+													+				7	
D. hookeri	+			+										+				7	
D. membranaceus										+	+		+	+			1	
D. strictus	+										+	+		+	+			15,14,1,2,13,12	
Dinochloa	+																		
D. sp.	+																	5	
Gigantochloa	+										+	+		+					
G. apus	+													+				9	
G. atter	+										+			+				5,16	
G. manggong	+											+		+				9	
Melocanna											+			+	+				
M. baccifera											+			+	+			1	
Nastus	+																		
N. reholttumianus	+																	5	
Phyllostachys	+			+										+					
P. mannii	+			+										+				7	
Schizostachyum	+													+					
S. brachycladum	+																	5	
S. lima	+																	5	
S. zollingeri	+													+				9	
Notes.

Ac Acaulosporaceae)

Am Ambisporaceae

Ar Archaeosporaceae

Cl Claroideoglomeraceae

Di Diversisporaceae

Gi Gigasporaceae

Gl Glomeraceae

Pa Paraglomeraceae

Aca Acaulospora

Ent Entrophospora

Ni Unidentified

Amb Ambispora

Cla Claroideoglomus

Div Diversispora

Gig Gigaspora

Scu Scutellospora

Glo Glomus

y Scl Sclerocystis

1 Jamaluddin & Turvedi (1997), 2 Jha et al. (2011), 3 Mafaziya, Wijewickrama & Madawala (2019), 4 Parkash, Handique & Dhungana (2019), 5 Kramadibrata (2011), 6 Weixin, Guangqian & Yulong (2013), 7 Das & Kayang (2010), 8 ((de Moura et al., 2019)), 9 Kramadibrata, Prastyo & Gunawan (2007), 10 Husein et al. (2022), 11 Verma & Arya (1998), 12 Ravikumar et al. (1997), 13 Muthukumar & Udaiyan (2006), 14 Bhattacharya et al. (2002), 15 Babu & Reddy (2010), 16 Mansir et al. (2021), 17 Qin et al. (2017a), 18 Qin et al. (2017b).

AMF composition associated with bamboos

In terms of the Glomeromycota diversity associated with bamboo species, only richness approximations, such as the number of reported species, and composition, such as the assignment of taxonomic identity to the group, were considered in the selected studies. Most studies taxonomically resolved the isolated morphological species from the soils associated with different bamboo species to the genus level, with some cases reaching a finer resolution to the species level. In some studies, although the presence of several morphological species within a given genus was recognized, the species was not reported, as in Mafaziya, Wijewickrama & Madawala (2019). Table 3 summarizes the genera and families of Glomeromycota fungi identified in soils associated with every bamboo species reported (grouped by genus), as well as the type of identification and the compartment from which the specimens were obtained. Figure 3 presents the bamboo genera and species associated with the Glomeromycota genera described in the selected documents, showing that Glomus and Acaulospora are the AMF genera associated with most of the bamboo genera (eight each) and species (25 and 23, respectively). Gigaspora followed with associations with six genera and 14 bamboo species, while Sclerocystis and Scutellospora recorded five genera and 12 bamboo species each, and Ambispora, Claroideoglomus, and Diversispora associated with two genera and two bamboo species each. Entrophospora and an unidentified Glomeromycota genus were reported to associate with only one bamboo species within a single bamboo genus.

Figure 3 Chord diagram of the genera of the phylum Glomeromycota associated with Bambusoideae.

On the left, the genera of Bambusoideae (on top) and the Glomeromycota genera with which they have been reported to associate. On the right, the genera of Glomeromycota (bottom) and the species of bamboo with which they have been reported to associate (top). Aca (Acaulospora), Ent (Entrophospora), Ni (Not identified), Amb (Ambispora), Cla (Claroideoglomus), Div (Diversispora), Gig (Gigaspora), Scu (Scutellospora), Glo (Glomus) and Scl (Sclerocystis). Bamboo genera: Phyll (Phyllostachys), Acti (Actinocladum), Bamb (Bambusa), Ceph (Cephalostachyum), Dend (Dendrocalamus), Dino (Dinochloa), Giga (Gigantochloa), Melo (Melocanna), Nast (Nastus), Schy (Schizostachyum). Bamboo species: A_ve (Actinocladum verticillatum), B_ar (Bambusa arundinacea), B_ba (Bambusa bambos), B_bl (Bambusa blumeana), B_bu (Bambusa burmanica), B_gr (Bambusa grandis), B_na (Bambusa nana), B_nu (Bambusa nutans), B_pa (Bambusa pallida), B_pe (Bambusa pervariabilis), B_po (Bambusa polymorpha), B_sp (Bambusa sp.), B_tu (Bambusa tulda), B_vu (Bambusa vulgaris), C_pe (Cephalostachyum pergracile), D_as (Dendrocalamus asper), D_ha (Dendrocalamus hamiltonii), D_ho (Dendrocalamus hookeri), D_me (Dendrocalamus membranaceus), D_st (Dendrocalamus strictus), D_sp (Dinochloa sp.), G_ap (Gigantochloa apus), G_at (Gigantochloa atter), G_ma (Gigantochloa manggong), M_ba (Melocanna baccifera), N_re (Nastus reholttumianus), P_ed (Phyllostachys edulis), P_ma (Phyllostachys mannii), S_br (Schizostachyum brachycladum), S_li (Schizostachyum lima), S_zo (Schizostachyum zollingeri).

Discussion

In addition to the collection and extraction of information related to the symbiotic association between Glomeromycota fungal taxa and Bambusoideae plants (which are described and gathered in the results), discoveries and descriptions of the relationships between symbionts in this association were also identified within the framework of this review from both the selected articles and those that were not necessarily included in the first part of this document. The central results of this review are summarized and discussed below, as well as the reported effects derived from the establishment of such arbuscular mycorrhizal symbiosis between fungi and bamboos from around the world.

AMF assemblages in soils associated with bamboo

Recent studies suggest that there are specific systems, such as those subject to high-impact management (e.g., inorganic fertilization, soil tillage, and understory removal) that induce alterations in the abundance and community composition of the AMF assemblage associated with bamboo, particularly Phyllostachys edulis (Qin et al., 2017a). Moreover, Qin et al. (2017a) also found that such management did not significantly affect the relative abundance of the AMF family Glomeraceae that was described to be the dominant group, which is consistent with reported communities in different ecosystems such as grasslands (Li et al., 2015), following the assertion that Glomeraceae is a disturbance-tolerant group (Chagnon et al., 2013). Acaulosporaceae, on the other hand, is not considered as such, and therefore it can be concluded that under these conditions, P. edulis is not an ideal host for this AMF family (Chagnon et al., 2013; Qin et al., 2017a). The most common families of AMF associated with bamboo were Acaulosporaceae and Glomeraceae, as compiled in Table 3 and Fig. 3.

Regarding systems derived from plantations, Jamaluddin & Turvedi (1997) reported the presence of five AMF genera (Glomus, Gigaspora, Acaulospora, Scutellospora, and Sclerocystis) in a bambusetum containing 13 bamboo species; however, despite mentioning some particular morphological species, they did not distinguish which bamboo sample was isolated. They also found that Bambusa nana presented the highest AMF colonization, followed by Bambusa vulgaris var. vittata, while the lowest colonization was obtained by Bambusa bambos, generating a gradient ranging from 80% to 33% in terms of mycorrhizal percentage (Jamaluddin & Turvedi, 1997). These results suggest different degrees of affinity between the plant and Glomeromycota symbionts, despite the fact that mycorrhizal symbiosis is well established in all cases.

Similarly, studies such as Parkash, Handique & Dhungana (2019) confirmed that Bambusa tulda, Bambusa pallida, Bambusa nutans, and Bambusa bambos form mycorrhizal associations with AMF after quantifying the percentage of root colonization by these fungal symbionts, with values in all cases exceeding 70%. In specific, within the rhizosphere of Bambusa bambos, they identified five morphological species of the genus Acaulospora (A. laevis, A. scrobiculata, A. lacunosa, A. mellea, and Acaulospora sp.), seven belonging to Glomus (G. clavisporum, G. reticulatum, G. macrocarpum, G. claroideum, G. pansihalos, G. geosporum), two in the genus Gigaspora (G. gigantea and Gigaspora sp.), and one Entrophospora sp. In the rhizosphere of B. tulda, they identified a total of four morphological species: Acaulospora foveata, Glomus clavisporum, G. albidum, and Scutellospora sp. Meanwhile, in Bambusa pallida, they reported eight morphological species, two Acaulospora (A. laevis and Acaulospora sp.) and six Glomus (G. macrocarpum, G. monosporum, G. geosporum, G. epigaeum, G. fasciculatum, and Glomus sp.); and in Bambusa nutans, they only recorded one species: Glomus epigaeum. Jha et al. (2011) found six species of Glomeromycota (Acaulospora scrobiculata, Glomus aggregatum, G. arborense, G. diaphanum, G. intraradices, and G. invermayanum) in the rhizospheric soil of Dendrocalamus strictus. Mansir et al. (2021) described the presence of three genera inside the roots of Gigantochloa atter, specifically Glomus, Gigaspora, and Acaulospora. In the rhizosphere of Bambusa sp., Husein et al. (2022) identified the presence of spores of the genera Glomus, Gigasporaceae, and Acaulospora, with several morphological species each, but their specific identities were not able to be determined. Additionally, in a study on populations of Bambusa bambos, Mafaziya, Wijewickrama & Madawala (2019) found 14 AMF morphotypes associated with the rhizospheric soil of the genera Glomus, Scutellospora, Gigaspora, and Acaulospora, with Glomus being the dominant genus and Acaulospora the least represented in terms of abundance.

Kramadibrata, Prastyo & Gunawan (2007) found that Acaulospora foveata and A. scrobiculata associated with Dendrocalamus asper and Gigantochloa apus, while Acaulospora tuberculata established associations with Bambusa vulgaris, Dendrocalamus asper, Schizostachyum zollingeri, Gigantochloa manggong, and Gigantochloa apus. In addition, Glomus etunicatum was associated with Bambusa vulgaris, Dendrocalamus asper, Gigantochloa manggong, and Gigantochloa apus; while Glomus fuegianum associated with B. vulgaris, D. asper, and Schizostachyum zollingeri. Glomus cf. formosanum and Glomus geosporum were associated with D. asper and G. apus. Glomus mosseae established associations with S. zollingeri and G. apus, and finally, Scutellospora calospora was associated with G. manggong. From the island of Sumba, Indonesia, Kramadibrata (2011) reported that Acaulospora foveata associated with Bambusa blumeana, Dinochloa sp., and Nastus reholttumianus, while Acaulospora scrobiculata associated with Bambusa blumeana, Schizostachyum brachycladum (green variety), and Nastus reholttumianus. Acaulospora tuberculata was found with Gigantochloa atter, Schizostachyum brachycladum (yellow variety), and Schizostachyum lima. Furthermore, Glomus etunicatum was associated with Gigantochloa atter, and finally, Glomus rubiforme with Gigantochloa atter.

In other studies, inoculants were prepared with isolated AMF identified as Glomus rosea, G. magnicaule, G. etunicatum, G. heterogama, G. maculosum, G. multicaule, Scutellospora nigra, and S. heterogama for application to bamboo. Babu & Reddy (2010) applied the inoculants to Dendrocalamus strictus and confirmed fungal colonization. Ravikumar et al. (1997) worked with independent inoculations and all possible combinations of Glomus aggregatum, G. fasciculatum, and G. mosseae on Dendrocalamus strictus. Under all circumstances, their roots were colonized with a colonization percentage ranging from 30% to 60%. This finding is similar to that of Verma & Arya (1998), in which five morphological species were isolated, including Acaulospora scrobiculata, Glomus intraradices, G. aggregatum, G. mosseae, and Scutellospora heterogama, from the rhizosphere of Dendrocalamus asper to develop two inoculants: one with the first species (I1) and the other with the remaining four (I2). Furthermore, they tested another inoculant (I3) from isolated teak spores, which contained Acaulospora scrobiculata, A. delicata, Gigaspora sp., G. ramisporophora, Glomus intraradices, G. geosporum, G. mosseae, G. etunicatum, and Scutellospora pellucida. These fungi established symbiotic associations with the bamboo species, which was confirmed by quantifying the percentage of colonization within the roots of D. asper. The statistical analyses showed that the maximum percentage of colonization was found in I3, followed by I1 and I2, respectively (Verma & Arya, 1998). In 2002, Dendrocalamus strictus was inoculated with three species of AMF, namely Glomus mosseae, G. fasciculatum, and Gigaspora margarita (Bhattacharya et al., 2002). Later, in 2011, Jha et al. (2011) inoculated Bambusa bambos and Dendrocalamus strictus with Acaulospora scrobiculata, A. mellea, Glomus aggregatum, G. cerebriforme, G. arborense, G. diaphanum, G. intraradices, G. etunicatum, G. fasciculatum, G. hoi, G. occultum, and Glomus sp., and all the fungi established an association with the roots of both bamboo species. Considering that the current veracity of fungal identity of spores present in commercial bio-inoculants is compromised (Vahter et al., 2023), the fact that most of the studies mentioned resorted to the use of trap cultures or direct isolation of Glomeromycota species to be inoculated provides some certainty regarding the accuracy of their results. It was frequently observed that Acaulospora and Glomus were among the most commonly isolated strains or fungal genera inoculated to different bamboo species, as these AMF genera are commonly found worldwide in a wide range of natural ecosystems including those altered by humans, such as agricultural systems (Davison et al., 2015; Oehl et al., 2017).

Effects of Glomeromycota on host bamboo plants

While the effects of symbiosis were not reported for all the bamboo species mentioned, we briefly summarized the physiological and morphological effects of Glomeromycota association with Bambusoideae described in the literature in Table 4. Some of them such as Dendrocalamus strictus were reported to significantly stabilize the upper layer of the soil due to their pachymorphic or sympodial rhizomes and enhance the leaf litter accumulation in response to associations with AMF (Ben-Zhi et al., 2005). Furthermore, seedlings of Dendrocalamus strictus that were inoculated with Glomus fasciculatum and G. mosseae (simultaneously) reached the greatest internodal distance of all treatments, followed by bamboo seedlings inoculated only with G. aggregatum (Ravikumar et al., 1997). The same study found that D. strictus rhizomes reached their maximum length when the inoculation contained only the G. aggregatum isolate, followed by the combination of G. aggregatum and G. mosseae. Total biomass production in D. strictus was also favored by mycorrhizal associations with Glomeromycota fungi, reaching maximum dry matter production when seedlings were inoculated with G. aggregatum and G. fasciculatum (Ravikumar et al., 1997).

Table 4 General physiological and morphological effects of AMF on bamboo.

Type of effect	Effect	References	
Morphological	Increased internodal distance	Ravikumar et al. (1997)	
	Increased rhizome length	Ravikumar et al. (1997)	
	Increased shot length	Jha et al. (2011)	
	Promoted lateral root branching and length	Bhattacharya et al. (2002)	
	Increased culm diameter	Weixin, Guangqian & Yulong (2013)	
	Increased total leaf area	Weixin, Guangqian & Yulong (2013)	
	Increased plant height	Verma & Arya (1998)	
Physiological	Improved nitrogen uptake	Muthukumar & Udaiyan (2006)	
	Improved phosphorus uptake	Verma & Arya (1998); Muthukumar & Udaiyan (2006); Babu & Reddy (2010); Jha et al. (2011); Weixin, Guangqian & Yulong (2013)	
	Improved calcium uptake	Babu & Reddy (2010)	
	Improved magnesium uptake	Babu & Reddy (2010)	
	Improved potassium uptake	Muthukumar & Udaiyan (2006); Babu & Reddy (2010); Weixin, Guangqian & Yulong (2013)	
	Increased biomass production	Ravikumar et al. (1997); Weixin, Guangqian & Yulong (2013)	
	Increased number of shoots	Weixin, Guangqian & Yulong (2013)	
	Reduced heavy metal translocation	Babu & Reddy (2010)	

In Bambusa bambos and Dendrocalamus strictus, phosphorus uptake, and shoot length increased significantly with all inoculated AMF (8 commercial formulations), except for Glomus sp. in D. strictus (Jha et al., 2011), indicating that effective AMF utilization can enhance productivity of these bamboo species in the region (Jha et al., 2011). Similarly, Bhattacharya et al. (2002) found that inoculating some AMF in D. strictus seedlings significantly promoted lateral branching (in number and length) of roots in this species, although no clarifications were made regarding the status of such fungi in the soil or even the percentage of mycorrhization, only reports of the morphological response of the seedling root system.

In terms of morphology, as mentioned above for species such as D. strictus, AMF inoculation causes differentiating effects on the structures of inoculated plants. For instance, inoculating G. mosseae and G. intraradices in Chenglu bamboo seedlings, a hybrid bamboo between Bambusa pervariabilis (as the female parent) and B. grandis (as the male parent), resulted in a significant increase in the number of shoots, culm diameter, and total leaf area of the inoculated plants compared to those not inoculated with AMF, as well as an improved uptake of P and K in inoculated plants, growth and biomass accumulation, making these AMF good candidates in efforts to increase the production of this hybrid bamboo (Weixin, Guangqian & Yulong, 2013).

In general, bamboo plants favor the establishment of mycorrhizal symbiosis given their fast growth which requires high nutrient requirements during the initial growth stages (Ravikumar et al., 1997). This explains the increase in the rhizosphere of several bamboo species after establishing a mycorrhizal symbiosis, a phenomenon that has specifically been described in Bambusa balcooa, Bambusa vulgaris var. vittata (known as green bamboo) and var. striata (known as yellow bamboo), Bambusa nutans and Dendrocalamus asper (Singh et al., 2020).

In Dendrocalamus strictus, inoculation of AMF along with Aspergillus tubingensis (an Ascomycota fungus) showed a synergistic effect on bamboo growth, nutrient uptake (such as P, K, Ca, and Mg), and reduction in heavy metal translocation to the plant (Babu & Reddy, 2010). Similar effects have also been observed in other plants that establish associations with AMF (Chen et al., 2007; Chen, Christie & Li, 2001). This is consistent with what Muthukumar & Udaiyan (2006) described, who performed a nursery experiment evaluating the effects of applying a Glomus aggregatum inoculum on promoting the growth of D. strictus plants in two different soil types (alfisol and vertisol). They found that root colonization percentages by G. aggregatum (reaching 55% and confirming the establishment of the association with this AMF species) were positively and significantly related to the dry weight of bamboo seedlings, and thus to the concentrations of N, P, and K in their tissues (roots, rhizome, and shoots) (Muthukumar & Udaiyan, 2006). In addition, Verma & Arya (1998) found that Dendrocalamus asper seedlings treated with AMF inocula extracted from their rhizospheric soil (I2 treatment) and one associated with Teak were significantly taller than those that did not receive treatments, and they also presented higher concentrations of P in their shoots, which they describe as an effect resulting from AMF-mediated improvements in the efficiency of capturing available P in the soil.

Effects of bamboo on AMF assemblages

Host plants are not passive actors during symbiosis, and AMF assemblages are not randomly distributed in patches of host plants but rather tend to associate with particular ecological groups (Davison et al., 2011). Considering this, Guo et al. (2023) described that the rhizospheric soil of bamboo, particularly Phyllostachys edulis and four of its forms (P. edulis f. tao kiang, P. edulis f. luteosulcata, P. edulis f. pachyloen and P. edulis f. gracilis) has more complex, longer, and interconnected fungal networks (including the phylum Glomeromycota) than those in non-rhizospheric soil, despite the fact that alpha diversity was significantly higher in the non-rhizospheric soil than in the rhizospheric soil, a pattern also observed in the abundance of Glomeromycota, which may be because the non-rhizospheric soil contained the original pool of species from which the plant root established selective associations with some of the AMF available in that compartment, thus reducing the richness and abundance of groups at the rhizospheric level (Bledsoe, Goodwillie & Peralta, 2020; Fiore-Donno et al., 2022). This contrasts with the results of Husein et al. (2022), who showed that the rhizosphere of Bambusa sp. has more abundance and diversity of Glomeromycota spores than those of other plants, such as Cichorium intybus and Pinus merkusii, which, although not compared to the non-rhizospheric soil of these species, gives indications of a possible affinity of Bambusa for selecting AMF, or the preference of AMF to associate with this bamboo species.

On the other hand, bamboos classified as runners (with leptomorph rhizomes), such as those of the genus Phyllostachys that are known for invading and quickly replacing neighboring forest cover, tend to increase the biomass of their associated AMF. This significantly changes the fungal assemblages in the soil, contributing to the formation of aggregates and carbon storage in the system (Xu et al., 2020; Qin et al., 2017a; Qin et al., 2017b). Likewise, when comparing a forest of Phyllostachys pubescens with surrounding forests, Qin et al. (2017a); Qin et al. (2017b) found that in the former, both AMF spore density and root colonization rate were significantly higher than in the latter, and that the former favored the presence of the Glomeraceae family but reduced that of the Acaulosporaceae and Archaeosporaceae families. Additionally, Jin et al. (2022) found that when extensive management is carried out in bamboo forests, particularly those of P. edulis (forests regularly harvested without any management), the abundance of AMF increases substantially leading to an increase in AMF respiration rates and a significant change in carbon cycling within bamboo forest ecosystems produced by AMF.

In contrast, Mafaziya, Wijewickrama & Madawala (2019) found no effects of an increase in the dominance of Bambusa bambos populations, in ecosystems where it is present, on the associated AMF assemblages. They concluded that, at least under the conditions of the study, the fungal community showed high resistance to changes in soil surface coverage, as well as resilience to the influence generated by the dominance of a single plant species.

Challenges ahead

Qin et al. (2017a), Qin et al. (2017b) and Jin et al. (2022) report molecular marker-based approaches to elucidate the composition of AMF assemblages in bamboo forest soils using primers described by Sato et al. (2005), that amplify part of the 18S region. These sequences can then be compared to the MaarjAM database using BLAST (Öpik et al., 2010) to assign taxonomic identities to the Glomeromycota groups in the sample. With similar approaches, Zhang et al. (2022) amplified the ITS region to evaluate the fungal community in the roots, soil, and aerial structures of Phyllostachys edulis, and Zhang et al. (2019) amplified the ITS2 region to investigate the rhizospheric community structure of P. edulis at the phylum level, identifying sequences using BLAST in the UNITE database. In those studies, they were able to identify groups within the Glomeromycota phylum, but the ITS region was proven not to be an optimal candidate for barcoding arbuscular mycorrhizal fungi (AMF) because the region alone is exceptionally variable and does not adequately resolve species, especially among closely related taxa. In addition, although the specific primers proposed by Sato et al. (2005) are promising in terms of coverage and specificity, Van Geel et al. (2014) and Stockinger, Krüger & Schüßler (2010) highlight that they are biased towards groups within the Glomeraceae family, as they do not detect any members of the Ambisporaceae, and only a few of the Claroideoglomeraceae and Paraglomeraceae within Glomeromycota, while even detecting groups in Basidiomycota and Chytridiomycota. This situation generates difficulties when comparing the results of studies on AMF diversity and composition, as there is little consistency regarding the target genes and primers used (Van Geel et al., 2014). While there is still no consensus region for AMF barcoding (Kolaříková et al., 2021), recent studies suggest that the ideal approach is to perform nested PCR approaches on different regions of the rRNA gene (Kolaříková et al., 2021). Tedersoo et al. (2022) also recommend that when performing taxonomic identification of a sequence using BLAST that may belong to Glomeromycota (if the selected primers allow it), the searches should preferably be made against the MaarjAM database, not UNITE, as the latter uses reads based on the ITS region, whose disadvantages have been previously mentioned.

Bamboo species such as Phyllostachys edulis have been described as promising in their roles of carbon sequestration through their mycorrhizal associations. This could lead to increases in planted areas or facilitate their expansion in ecosystems through the application of Glomeromycota bio-inoculants. However, it is imperative to pay attention to the ecological risks this may also entail in terms of negative effects on plant diversity and other soil microorganisms in the colonized areas (Qin et al., 2017a; Qin et al., 2017b). Intensive management practices traditionally applied to these bamboo species reduces soil pH, facilitates the hyper accumulation of available N, P, and K, and promotes soil aggregation loss and erosion, which leads to a significant reduction in AMF biomass as well as alterations in soil assemblage diversity (Xu, Jiang & Xu, 2008; Shinohara & Otsuki, 2015; Qin et al., 2017a; Liu et al., 2011). This situation can be avoided by promoting responsible and sustainable use and management of alternative practices, such as reduced tillage and organic amendments (Qin et al., 2017a).

As for publications addressing the establishment of Glomeromycota-bamboo symbiotic associations, although some report evidence of AMF colonization in roots of species such as Dendrocalamus strictus (Das et al, 2021), they do not include any other evidence in terms of spore identification or genetic material extraction from any of the different compartments where they are found (soil and/or the root of the host plant). As a result, the information conveyed is poor or incomplete, at least within the framework of the objectives of this work.

Moreover, in some cases, the term ‘bamboo’ is used in a very generic way or even as if constituting a taxonomic rank by itself. This has led to several studies maintaining an ambiguous discourse when discussing the potential of bamboo (in ecological or economic terms) or even of their associated AMF species (i.e., Toh et al., 2018; Patra et al., 2021; Priya et al., 2014; Mafaziya, Wijewickrama & Madawala, 2019), resulting in a loss of informative value. Additionally, as reported by Debnath et al. (2015), several AMF genera were isolated from different sampling sites (Acaulospora, Ambispora, Diversispora, Funneliformis, Glomus, Paraglomus, Rhizophagus, and Sclerocystis), but the associated bamboo species was not clarified, nor is it considered that they may be working with other species, which generates noise in the analyses. Rather, studies must clarify which species of bamboo is being referred to, since as evidenced throughout this document, the nature of the symbiosis, its effects, and the potentials of each species of bamboo are, to some degree, specific.

In addition, it is surprising that in tropical America, a continent where this group of plants is ecologically, economically, and culturally important and diverse (i.e., Guadua spp., specifrically Guadua angustifolia in terms of its uses, applications, and distribution range) (Cruz-Armendáriz, Ruiz-Sanchez & Reyes-Agüero, 2023; Akinlabi, Anane-Fenin & Akwada, 2017; Clark, Londoño & Ruiz-Sanchez, 2015), associations with AMF are so absent in the scientific literature, at least in publications from indexed and/or peer-reviewed journals. This phenomenon may be the result of the ease of producing “gray literature” in terms of speed and low cost, or even the difficulties involved in submitting and publishing a scientific document (Corlett, 2010). This greatly hinders access to information and the construction of knowledge on environmental and social issues such as those exposed in this document. Therefore, the publication of studies focused on this association is the first step in supporting efforts aimed to propagate bamboos using AMF in various regions of the world, as proposed by de Moura et al. (2019). With adequate knowledge of mycorrhizal fungal diversity in the rhizosphere of bamboo species and molecular approaches for analyzing roots and soils, the development of mycorrhizal inoculation programs is guaranteed to make bamboo available as a sustainable resource, while expanding the spectrum of possibilities in terms of the applicability of these species for ecological restoration and the fight against socio-environmental problems (Das & Kayang, 2010; de Moura et al., 2019).

Finally, it is important to note that although this document did not modify the names of the taxa assigned to Glomeromycota and Bambusoideae reported in the reviewed documents, taxonomic reorganizations and changes in names or categories have occurred in several cases (i.e., Błaszkowski et al., 2022), with some being assigned as synonyms of a more accepted name, for example. In some cases, even the reported names were fundamentally incorrect, not stemming from any synonymy confusion. Therefore, it is relevant to keep up to date with nomenclatural changes associated with the study groups being investigated to facilitate research and communication both within and outside the academic community. This can be done through specialized databases such as Index Fungorum (http://www.indexfungorum.org/, Species Fungorum (https://www.speciesfungorum.org/, and MycoBank (http://www.mycobank.org/ for Glomeromycota fungi, and literature such as the freely accessible book “World Checklist of Bamboos and Rattans” by Vorontsova et al. (2016) for bamboos. The current status of the names of Glomeromycota fungi and Bambusoideae bamboos are summarized in Tables 5 and 6, respectively.

Table 5 Names of the Glomeromycota taxa mentioned in the text, and the current status of these names according to Index Fungorum, Species Fungorum, and MycoBank.

Name mentioned in text	Current/accepted name (March 2023)	
Acaulospora	Acaulospora Gerd. & Trappe	
A. scrobiculata	Acaulospora scrobiculata Trappe	
A. foveata	Acaulosporafoveata Trappe & Janos	
A. laevis	Acaulospora laevis Gerd. & Trappe	
A. mellea	Acaulospora mellea Spain & N.C. Schenck	
A. lacunosa	Acaulospora lacunosa J.B. Morton	
A. tuberculata	Acaulospora tuberculata Janos & Trappe	
A. delicata	Acaulospora delicata C. Walker, C.M. Pfeiff. & Bloss	
A. rehmii	Acaulospora rehmii Sieverd. & S. Toro	
A. cavernata	Acaulospora cavernata Blaszk	
Ambispora	Ambispora C. Walker, Vestberg & A. Schüssler	
A. leptoticha	Ambispora leptoticha (N.C. Schenck & G.S. Sm.) C. Walker, Vestberg & A. Schüssler	
Claroideoglomus	Entrophospora R.N. Ames & R.W. Schneid.	
Diversispora	Diversispora C. Walker & A. Schüßler	
Entrophospora	Entrophospora R.N. Ames & R.W. Schneid.	
Gigaspora	Gigaspora Gerd. & Trappe	
G. margarita	Gigaspora margarita W.N. Becker & I.R. Hall	
G. gigantea	Gigaspora gigantea (T.H. Nicolson & Gerd.) Gerd. & Trappe	
Glomus	Glomus Tul. & C. Tul.	
G. mosseae	Funneliformis mosseae (T.H. Nicolson & Gerd.) C. Walker & A. Schüßler	
G. rosea	Gigaspora rosea T.H. Nicolson & N.C. Schenck	
G. magnicaule	Glomus magnicaule I.R. Hall	
G. multicaule	Glomus multicaule Gerd. & B.K. Bakshi	
G. etunicatum	Entrophospora etunicata (W.N. Becker & Gerd.) Błaszk., Niezgoda, B.T. Goto & Magurno	
G. heterogama	Dentiscutata heterogama (T.H. Nicolson & Gerd.) Sieverd., F.A. Souza & Oehl	
G. maculosum	Glomus maculosum D.D. Mill. & C. Walker	
G. aggregatum	Rhizoglomus aggregatum (N.C. Schenck & G.S. Sm.) Sieverd., G.A. Silva & Oehl	
G. fasciculatum	Rhizoglomus fasciculatum (Thaxt.) Sieverd., G.A. Silva & Oehl	
G. intraradices	Rhizoglomus intraradices (N.C. Schenck & G.S. Sm.) Sieverd., G.A. Silva & Oehl	
G. clavisporum	Glomus clavisporum (Trappe) R.T. Almeida & N.C. Schenck	
G. reticulatum	Glomus reticulatum Bhattacharjee & Mukerji, Sydowia	
G. macrocarpum	Glomus macrocarpum Tul. & C. Tul.	
G. claroideum	Entrophospora claroidea (N.C. Schenck & G.S. Sm.) Błaszk., Niezgoda, B.T. Goto & Magurno	
G. pansihalos	Halonatospora pansihalos (S.M. Berch & Koske) Błaszk., Niezgoda, B.T. Goto & Kozłowska	
G. geosporum	Funneliformis geosporum (T.H. Nicolson & Gerd.) C. Walker & A. Schüßler	
G. clavisporum	Glomus clavisporum (Trappe) R.T. Almeida & N.C. Schenck	
G. monosporum	Funneliformis monosporus (Gerd. & Trappe) Oehl, G.A. Silva & Sieverd.	
G. albidum	Paraglomus albidum (C. Walker & L.H. Rhodes) Oehl, F.A. Souza, G.A. Silva & Sieverd.	
G. cerebriforme	Glomus cerebriforme McGee	
G. arborense	Glomus arborense McGee	
G. hoi	Simiglomus hoi (S.M. Berch & Trappe) G.A. Silva, Oehl & Sieverd.	
G. diaphanum	Oehlia diaphana (J.B. Morton & C. Walker) Błaszk., Kozłowska, Niezgoda, B.T. Goto & Dalpé	
G. occultum	Paraglomus occultum (C. Walker) J.B. Morton & D. Redecker	
G. rubiforme	Glomus rubiforme (Gerd. & Trappe) R.T. Almeida & N.C. Schenck	
G. epigaeum	Diversispora epigaea (B.A. Daniels & Trappe) C. Walker & A. Schüßler	
G. tortuosum	Sieverdingia tortuosa (N.C. Schenck & G.S. Sm.) Błaszk., Niezgoda & B.T. Goto	
G. constrictum	Septoglomus constrictum (Trappe) Sieverd., G.A. Silva & Oehl	
Name mentioned in text	Current/accepted name (March 2023)	
Sclerocystis	Sclerocystis Berk. & Broome	
Scutellospora	Scutellospora C. Walker & F.E. Sanders	
S. nigra	Dentiscutata nigra (J.F. Redhead) Sieverd., F.A. Souza & Oehl	
S. heterogama	Dentiscutata heterogama (T.H. Nicolson & Gerd.) Sieverd., F.A. Souza & Oehl	
S. calospora	Scutellospora calospora (T.H. Nicolson & Gerd.) C. Walker & F.E. Sanders	
S. pellucida	Cetraspora pellucida (T.H. Nicolson & N.C. Schenck) Oehl, F.A. Souza & Sieverd.	

Table 6 Names of the Bambusoideae taxa mentioned in the text, and the current status of these names according to Vorontsova et al. (2016).

Name mentioned in text	Current/accepted name (March 2023)	
Actinocladum	Actinocladum McClure ex Soderstr	
Actinocladum verticillatum	Actinocladum verticillatum (Nees) McClure ex Soderstr.	
Bambusa	Bambusa Schreb.	
Bambusa arundinacea	Bambusa bambos (L.) Voss	
Bambusa bambos	Bambusa bambos (L.) Voss	
Bambusa blumeana	Bambusa tuldoides Munro	
Bambusa burmanica	Bambusa burmanica Gamble	
Bambusa grandis	Bambusa grandis (Q.H.Dai & X.L.Tao) Ohrnb.	
Bambusa nana	Bambusa multiplex (Lour.) Raeusch. ex Schult.f.	
Bambusa nutans	Bambusa nutans Wall. ex Munro	
Bambusa pallida	Bambusa pallida Munro	
Bambusa pervariabilis	Bambusa pervariabilis McClure	
Bambusa polymorpha	Bambusa polymorpha Munro	
Bambusa tulda	Bambusa tulda Roxb.	
Bambusa vulgaris	Bambusa vulgaris Schrad. ex J.C.Wendl.	
Bambusa sp.	Bambusa sp.	
Cephalostachyum	Cephalostachyum Munro	
Cephalostachyum pergracile	Schizostachyum pergracile (Munro) R.B.Majumdar	
Dendrocalamus	Dendrocalamus Nees	
Dendrocalamus asper	Dendrocalamus asper (Schult.f.) Backer	
Dendrocalamus hamiltonii	Dendrocalamus hamiltonii Nees & Arn. ex Munro	
Dendrocalamus hookeri	Dendrocalamus hookeri Munro	
Dendrocalamus membranaceus	Dendrocalamus membranaceus Munro	
Dendrocalamus strictus	Dendrocalamus strictus (Roxb.) Nees	
Dinochloa	Dinochloa Buse	
Dinochloa sp.	Dinochloa sp.	
Gigantochloa	Gigantochloa Kurz ex Munro	
Gigantochloa apus	Gigantochloa apus (Schult.f.) Kurz	
Gigantochloa atter	Gigantochloa atter (Hassk.) Kurz	
Gigantochloa manggong	Gigantochloa manggong Widjaja	
Melocanna	Melocanna Trin.	
Melocanna baccifera	Melocanna baccifera (Roxb.) Kurz	
Nastus	Nastus Juss.	
Nastus reholttumianus	Nastus reholttumianus Soenarko	
Phyllostachys	Phyllostachys Siebold & Zucc.	
Phyllostachys edulis	Phyllostachys edulis (Carrière) J.Houz.	
Phyllostachys mannii	Phyllostachys mannii Gamble	
Schizostachyum	Schizostachyum Nees	
Schizostachyum brachycladum	Schizostachyum brachycladum (Kurz ex Munro) Kurz	
Schizostachyum lima	Schizostachyum lima (Blanco) Merr.	
Schizostachyum zollingeri	Schizostachyum zollingeri Steud.	

Conclusions

This study presents the first systematic review of the current state of knowledge surrounding arbuscular mycorrhizal symbiosis established between plants and fungal taxa within Bambusoideae and Glomeromycota worldwide. Asia, particularly India and China, stood out as the continent and countries with the most (and almost all) studies on the topic of review. The present review also highlights the need to compile information and build knowledge around a wider range of bamboo species and the symbiotic interactions they form or can potentially form with AMF, as these topics remain incipient and poorly explored particularly with respect to the biology and ecology of mycorrhizal symbiosis in Bambusoideae. Here, we compiled a list of the bamboo species that have been evaluated, described, or evidenced to establish mutualistic associations with arbuscular mycorrhizal fungi, including 31 species within 10 botanical families that associate with a considerable diversity of Glomeromycota genera and families, 17 and eight, respectively.

Supplemental Information

Supplemental Information 1 Systematic Review and/or Meta-Analysis Rationale

Click here for additional data file.

Supplemental Information 2 PRISMA Checklist

Click here for additional data file.

The authors wish to thank Eduardo Ruíz-Sánchez for his valuable contribution.

Additional Information and Declarations

Competing Interests

Author Contributions

Data Availability

The authors declare there are no competing interests.

Juan José Sánchez-Matiz conceived and designed the experiments, performed the experiments, analyzed the data, prepared figures and/or tables, authored or reviewed drafts of the article, and approved the final draft.

Lucia Ana Díaz-Ariza conceived and designed the experiments, performed the experiments, analyzed the data, prepared figures and/or tables, authored or reviewed drafts of the article, and approved the final draft.

The following information was supplied regarding data availability:

This article is a literature review.

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
