# Peer review of "Glomeromycota associations with bamboos (Bambusoideae) worldwide, a qualitative systematic review of a promising symbiosis"

_PeerJ, doi:10.7717/peerj.16151_

## Round 0.1 · original submission · Minor Revisions

Dear Dr. Díaz-Ariza,

Thank you for your submission to PeerJ.

Your submission had reviewed by two experts in your research. It took little bit longer than expected and thank you for your patience and understanding.

Both reviewers have recommended minor revisions but they also suggested several comments to improve your manuscript. I also have a few minor edits in the attached pdf file. Based on the reviewers’ recommendations and my own assessment, I ask you to read all comments provided by the reviewers, revise it, and then submit to PeerJ for publication at a suitable time.

In addition, the introduction, results, and discussion must be concise and precise. All figures/tables qualities need to improve and must be self-explanatory. Importantly, thorough English is necessary for clarity.

With kind regards,

Tika Adhikari

Academic Editor
PeerJ Life & Environment

Reviewer 1 ·

Basic reporting

The written English is acceptable, and the structure, figures, tables are professional.

Experimental design

no comment

Validity of the findings

no comment

Additional comments

The manuscript presents a systematic review of the current state of knowledge surrounding arbuscular mycorrhizal symbiosis established between plants and fungal taxa within Glomeromycota and Bambusoideae worldwide. The analysis is useful for people to understand the relationships between Glomeromycota and Bambusoideae, which could be helpful for the further management practices aimed at improving plant productivity and reducing ecological risks. I have only several detailed suggestions on this manuscript.
1. in the Introduction part, the authors reviewed the research results about both bamboo and AM fungi, but why the authors want to reveal the relationships between them? I suggest the authors to describe the importance of AMF on bamboo growth or other ecological functions.
2. the discussion was too long, and some parts can not be distinguished from each other. For example, in the “effects of Glomeromycota on host bamboo plants” part, there are two subject part, but both of them discussed about the growth of bamboo. Thus, I suggest to combine these two subject together and simplify this part.
3. Line 470, this paragraph seems not the effects of bamboo on AMF assemblages.
4. the authors should simplify the “challenges ahead” and the “conclusion” part, especially for the conclusion part, to make readers get the point quickly and clearly.
5. one suggestion for the table, if the table is separated in two or more pages, I suggest to add the header cell in each page.

Reviewer 2 ·

Basic reporting

In this manuscript the authors summarize the current existing literature on the AMF species associated to bamboo species and genera worldwide, and address on the gap the of knowledge remaining on this subject. They describe the frameworks used in the different studies and the most common genera of AMF found in the different bamboo genera and species.

I found the manuscript interesting, and showing the need to develop more studies in the field of bamboo AMF interactions taxonomy. The manuscript is in general well written, but I have some suggestions that could improve its understanding.

Introduction: It would be useful to add a some more details on the bamboo characteristics, specially of their rhizome systems and on the fact that it is considered as an invasive plant in some occasions.
In line 39. What does prolonged intervals mean?
Also, It is not clear for me if there are any records of this family in the remaining continents that were excluded from this study.
I suggest that the last sentence from Introduction: "The stimulation of their growth and development can be achieved through the use of beneficial microorganisms such as AMF” should be linked with the importance of the objectives of your work.

Experimental design

Experimental design is clear, research question is well defined.
Figure 1 have two phrases to be translated: Fuera de contexto and Revisiones Narrativas.

Validity of the findings

I would find of interest to evaluate if there is any correlation between bamboo taxonomic groups inside the Bambusoideae subfamily and their AMF diversity, or their preference for certain AMF families.
Even though in Figure 3 it can be seen which AMF genera relate to which Bamboo species, it would be relevant to evaluate if there is any pattern in AMF preference regarding diffeerent bamboo tribes, or differences among herbaceous and woody bamboos.
Also, when it is mentioned the effects of AMF on plant growth and nutrient uptake, a metanalysis among these studies, or at least a summarised table of results would be helpful to overview the general effects the AMF had on these plants so far.

Additional comments

Lines 49/51. Sentence should be rewritten. Maybe: ``the areas where these plants are most cultivated and consumed are concentrated in Asia, Africa and America (cites), where there is also the greater diversity (cite).”
Line 56. Replace: “constitute diverse niches for microorganisms” for: “constitute niches for diverse microorganisms”
Line 56/57. Rewrite this sentence, maybe: Among the multiple interactions between plants and organisms, stand out the ones established with the phylum Glomeromycota
Line 74. Should be: "mainly" in the form of carbohydrates
Line 87. “It is required” instead of: it requires.
Line 96 add “of literature” in: screening, eligibility, and inclusion of literature
Line 112. Replace: The reference lists of the selected literature and articles were used to find additional articles, by: The reference list of the selected literature was used to exclude additional articles
Line 147. Remove initials in the cites.
Line 156. Did you specifically eliminated articles mentioning species of bamboo and AMF from other continents? Is bamboo even present in these other continents?
Line 159. I suggest “Bambusoideae taxa reported from each country and the literature references are presented in Table 1.”
Lines 180-187. Use the past tense in all the paragraph.
Line 187. What do you mean by edaphic and biotic variables of bamboo?
Line 197. It should be mentioned what kind of AMF or soil they used in this bambusetum, natural soils? from where? Did they extract any other conclusion from these results?
Line 198. Sampling sites from where?
Line 204. Add in this sentence: Significant differences in colonization rates "among the two plant species".
Cut the sentence in two: Also, Das & Kayang (2010) described that Bambusa tulda exhibits an Arum-type AMF colonization….
Explain shortly what does it imply or why it would be of interest this difference in type of colonization.
Line 208. I suggest to rewrite: Another case (Fukuchi et al. 2011), described how the Japanese bamboo species Sasa senanensis establishes arbuscular mycorrhizal associations with Glomeromycota; however, it did not report the fungal taxonomy.
Line 220. Was "the" type of tools used for the identification of Glomeromycota
Line 230. Replace by: compartment analyzed in all studies
Line 245. Replace by: the most bamboo genera with most of the bamboo genera.
Table 3. I suggest to add vertical borders in the table separating between Glomeromycota families to make an easier visualization of results.
Line 295. Rewrite ths sentence to be clearer, as the table 3 does not reflect that “ there are specific systems, such as those subject to high-impact management, where the composition of the AMF assemblage associated with Phyllostachys edulis was dominated by the Glomeraceae family in terms of abundance in sequence numbers (Qin et al., 2017a). Please clarify this.
But it could be said that bamboo species grown in high impact managements are related to Glomeraceae families, or is it P. edulis the only observed case?
Line 299. This fungal family instead of the fungal family.
Line 298. Change: “However” in this sentence as it is not in opposition to previous statement. It is not clear to me what this study found, maybe rewrite to make it clearer.
Line 302. Replace "Acaulosporaceae" by "this AMF family".
Lines 303/310. Was this percentage of colonization found in different species of bamboo correlated with their taxonomic grouping, or growing pattern?
Line 385. What do you mean by: the nature of their root system?
Line 430. Does “(2)” refer to the second inoculant treatment mentioned above (I2) please clarify.
Line 452. Is it higher diversity?
Line 452/455. This sentence is not clear to me, please rewrite.
Line 517. Which type of management?
Line 522. What do you mean by intensive management strategies traditionally applied to this bamboo species and why should it be promoted?
Line 585. “Bambusoideae and Glomeromycota (respectively) worldwide”
Line 589. Delete “filtering and” in “as the filtering and selection process filtered documents classified as gray literature…:”
Line 769. Correct citing format.

---

## Round 0.2 · accepted · Accept

Dear Dr. Díaz-Ariza:

Thank you for submitting your manuscript to PeerJ for publication. Due to summer vacations and travels, it took little bit longer to complete review process than expected.

Finally, I am pleased to accept your manuscript. Congratulations.

Best regards,

Sincerely,

Tika Adhikari